# Integrated Analysis of Small RNA, Transcriptome, and Degradome Sequencing Reveals the Water-Deficit and Heat Stress Response Network in Durum Wheat

**DOI:** 10.3390/ijms21176017

**Published:** 2020-08-21

**Authors:** Haipei Liu, Amanda J. Able, Jason A. Able

**Affiliations:** School of Agriculture, Food & Wine, Waite Research Institute, The University of Adelaide, Urrbrae, SA 5064, Australia; amanda.able@adelaide.edu.au (A.J.A.); jason.able@adelaide.edu.au (J.A.A.)

**Keywords:** next-generation sequencing, microRNAs, transcriptome, degradome, durum wheat, water-deficit stress, heat stress, epigenetics, cereal crop improvement

## Abstract

Water-deficit and heat stress negatively impact crop production. Mechanisms underlying the response of durum wheat to such stresses are not well understood. With the new durum wheat genome assembly, we conducted the first multi-omics analysis with next-generation sequencing, providing a comprehensive description of the durum wheat small RNAome (sRNAome), mRNA transcriptome, and degradome. Single and combined water-deficit and heat stress were applied to stress-tolerant and -sensitive Australian genotypes to study their response at multiple time-points during reproduction. Analysis of 120 sRNA libraries identified 523 microRNAs (miRNAs), of which 55 were novel. Differentially expressed miRNAs (DEMs) were identified that had significantly altered expression subject to stress type, genotype, and time-point. Transcriptome sequencing identified 49,436 genes, with differentially expressed genes (DEGs) linked to processes associated with hormone homeostasis, photosynthesis, and signaling. With the first durum wheat degradome report, over 100,000 transcript target sites were characterized, and new miRNA-mRNA regulatory pairs were discovered. Integrated omics analysis identified key miRNA-mRNA modules (particularly, novel pairs of miRNAs and transcription factors) with antagonistic regulatory patterns subject to different stresses. GO (Gene Ontology) and KEGG (Kyoto Encyclopedia of Genes and Genomes) enrichment analysis revealed significant roles in plant growth and stress adaptation. Our research provides novel and fundamental knowledge, at the whole-genome level, for transcriptional and post-transcriptional stress regulation in durum wheat.

## 1. Introduction

Water-deficit and heat are significant environmental threats that negatively impact crop production and quality, ultimately putting world food security at risk [1]. Durum wheat (2*n* = 28, AABB, *Triticum turgidum durum* L.) is the most widely cultivated tetraploid and is used for its high grain quality and versatility in making products, such as pasta [2,3]. Durum wheat is mainly grown in semi-arid Mediterranean environments where water scarcity and high temperature often occur [4,5]. Improving crop resilience to water-deficit and heat stress has become a major target for breeding programs.

For cereal crops, grain production largely depends on the reproductive processes, such as flowering and grain fill, both of which are highly sensitive to abiotic stress [1,6]. Crop plants have developed dynamic response mechanisms at the morphological, physiological, and biochemical levels to mitigate the negative effects of abiotic stress and to ensure reproductive success. To cope with water deficiency, the physiological adaptation in wheat species include mechanisms to maintain plant water status, such as limiting water loss (through stomatal closure or reduced leaf area) and maximizing water uptake (increased lateral roots or deeper root systems) [7,8,9,10]. Other tolerance strategies to reduce cellular damage caused by water deficiency include osmotic adjustments and protective enzyme activities to enhance reactive oxygen species (ROS) scavenging [11,12]. To cope with heat stress, physiological and metabolic adaptations in wheat species include increased production of chaperone proteins like heat shock proteins (HSPs); the activation of antioxidant defense system to minimize heat-induced oxidative damage; changes in cellular structures, such as membrane fluidity adjustment to enhance membrane stability [13,14,15]. In Mediterranean conditions, such as those typically found across the Australian wheat belt, water-deficit stress usually starts during spring before anthesis, and water deficiency often continues to intensify through grain filling [8,9]. Short periods of heat stress during the post-anthesis period are also prevalent in Australian field environments [16,17]. In Australian durum wheat genotypes, water-deficit stress applied from the booting stage until harvest had caused a reduction in grain number per plant by up to 64.6%, and reduction in grain yield per plant by up to 61.6% [9]. The negative impacts of reproductive-stage water deficiency on grain number are predominantly linked to the decrease of photosynthetic activity, which causes insufficient source supply to the reproductive tissues, eventually leading to floret abortion and failed grain formation [9,18]. In Italian durum wheat cultivars, just a small increase in mean minimum air temperature post-anthesis caused grain yield reduction by up to 22.6% [19]. Significant genotypic differences in yield performance were observed between modern and traditional cultivars, mainly attributed to differences in their leaf traits, such as net photosynthesis rate and transpiration rate [19]. Although water-deficit stress and high temperature are commonly coupled in the field, their combined effects can be quite different from their individual impact [1,8]. Therefore, to build crop resilience, it is crucial to investigate the effects of water-deficit stress and heat stress independently and combined, in a systematic manner. Moreover, the additive influence of reproductive-stage water-deficit and heat stress is genotype-dependent. In our previous study [8], Australian genotypes with varying stress tolerance levels exhibited contrasting responses to water-deficit and heat stress—not only for yield components but also for quality traits, such as grain protein content, and physiological traits, such as leaf chlorophyll content. This genotypic diversity in stress adaptation is underpinned by transcriptional and post-transcriptional regulatory molecular mechanisms that could be harnessed to breed for crop resilience [20]. However, currently, the stress regulatory molecular network in durum wheat is not well understood.

Recent biotechnology advances like next-generation sequencing (NGS) have significantly fast-tracked the identification of novel regulators with significant potential for crop improvement. Plant microRNAs (miRNAs), a type of small non-coding RNAs (sRNAs), are promising candidates for improving crop resilience [21,22,23]. As epigenetic regulators, miRNAs provide more efficient and precise regulation of gene expression in a sequence-specific manner [24,25]. Plant miRNAs can rapidly respond to different developmental and environmental signals and subsequently suppress the expression of their targets via mRNA cleavage or transcriptional inhibition [23,26,27,28]. These miRNA-target modules play significant roles in biological processes, such as signal transduction, nutrient transport, reproductive development, organ formation, and particularly, stress adaptation [29,30,31,32,33,34,35,36]. In crop plants, upon abiotic stress, miRNAs can quickly reprogram the expression of downstream genes that tightly regulate adaptive physiological traits, such as altered flowering timing and increased root water uptake. In wheat species, the target repertoire of stress-responsive miRNA includes many key genes involved in the molecular mechanism underlying stress tolerance (e.g., hormone biosynthetic enzymes, nucleic acid binding proteins, ROS scavengers, nutrient metabolic enzymes, and transporter proteins). Widely-known examples include modules like miR167 and ARF (auxin response factors), regulating auxin signaling; miR156 and SPL (Squamosa promoter binding protein-like) transcription factors, regulating flowering time; miR398 and superoxide dismutases, regulating ROS scavenging; miR164 and HSPs, regulating cellular protein conformation [37,38,39,40]. Studies in cereals have also reported contrasting regulatory patterns of miRNA-mRNA modules between stress-tolerant and -sensitive genotypes [18,41,42,43]. The molecular functions of these modules have been linked to physiological and yield performance, confirming that miRNAs are promising candidates for crop improvement. Previous reports on durum wheat used the sRNA-seq technique to investigate miRNA populations under nitrogen stress [44,45], water-deficit stress at the flowering time [46], and drought stress at the seedling stage [47]. A study with Italian cultivars also looked at miRNAs in response to short-term drought stress (2–3 days) and heat stress (3 h) in young seedlings [41]. However, no research has been conducted to elucidate miRNA functions specific to stress type, reproductive stage, and by using the durum wheat miRNAome exposed to individual and combined water-deficit and heat stress at different reproductive stages, among genotypes with contrasting tolerance levels. Previous research on durum wheat miRNA discovery also mainly used Italian cultivars [48,49], and there has been limited information on miRNA populations in Australian durum wheat germplasm. New knowledge gained on the miRNA population diversity in Australian genotypes will provide new opportunities for researchers or breeding programs interested in utilizing the genetic/epigenetic diversity in Australian durum wheat germplasm. Moreover, genomes of several cereal species are well known for their high level of complexity (polyploidy) and large size [50,51]. Lack of a high-quality reference genome has hindered epigenetic research in durum wheat, until recently where the first full-genome assembly (~10.45 GB) of durum wheat was reported [52], thus making it possible to further decrypt and harness new knowledge from NGS datasets. 

Apart from sRNA-seq, mRNA-seq (or transcriptome-seq) is another powerful high-throughput method for investigating stress regulatory networks [53,54,55]. Transcriptome studies in bread wheat [56,57] discovered a significant number of *Triticum aestivum* gene families involved in water-deficit and heat stress responses, such as receptor-like kinases, mitogen-activated protein kinase cascades, and transporter proteins. In durum wheat, by using cDNA-AFLP or array-based approaches, studies have identified drought- and heat-responsive genes involved in epigenetic regulation, ABA signaling, and senescence [58,59]. However, NGS-based transcriptome datasets have not been available in durum wheat to decipher the water-deficit/heat stress networks. Additionally, RNA-seq technology can be used for parallel analysis of RNA ends, i.e., degradome sequencing. Degradome-seq globally captures and sequences the uncapped 5′ ends of cleaved mRNA sequences, to identify mRNA degradation sites induced by miRNAs [60,61]. It is currently the only high-throughput NGS approach that identifies mRNAs targeted by miRNAs on the genome-wide scale. A significant number of miRNA-mRNA pairs have been validated via this method in bread wheat, rice, barley, and maize [62,63,64,65], but such research has not been conducted in durum wheat. Moreover, due to the gene silencing effect of miRNAs, miRNA-mRNA pairs often show spatiotemporal expression that is antagonistically regulated (i.e., up-regulation of miRNAs inversely correlated with down-regulation of mRNAs, and vice versa). Genome-wide analysis of such co-expression patterns is an important indicator that links the biological interactions of miRNA-target modules. This could only be achieved with a multi-omics analysis that collectively analyzes datasets of sRNA-seq, mRNA-seq, and degradome-seq at the same time. Previous research has successfully employed such a multi-omics approach in species like rice, maize, and chickpea [66,67,68,69]. However, it has not been previously undertaken in durum wheat. 

Here, we performed the first multi-omics analysis of the durum sRNAome, transcriptome, and degradome using NGS, aiming to systematically elucidate the molecular networks in response to single and combined water-deficit and heat stress. The number of sRNA libraries produced (a total of 120) was unprecedented in crop epigenetic research. We identified 523 durum wheat miRNAs, with 55 of them being novel. Transcriptome-seq discovered a total of 49,436 genes, with differentially expressed genes (DEGs) associated with hormone signaling, photosynthesis, and metabolic processes. Degradome-seq discovered over 100,000 miRNA-guided cleavage sites, many of which were unconfirmed before the current study. Newly identified mRNA targets included a significant number of transcription factors, protein kinases, and hormone regulators. Multi-omics analysis identified key miRNA-target pairs that were antagonistically regulated. Stress-responsive differentially expressed miRNAs (DEMs) and DEGs with genotypic patterns were involved in hormone homeostasis, transmembrane transport, and signaling transduction pathways. The results provided a substantial amount of novel and fundamentally important information of the durum wheat stress-response networks. Newly-discovered DEMs and DEGs provided valuable genetic and epigenetic resources for breeding programs aimed at enhancing varietal improvement to specific stresses. 

## 2. Results

### 2.1. Conserved and Novel miRNAs Identified in sRNA-seq

Sequencing of 120 sRNA libraries (Appendix A) generated over 1.20 billion raw reads representing 241.85 million unique reads. After trimming and processing, 342.30 million clean sRNA reads were obtained. The size distribution of sRNA reads exhibited a classic two-peak pattern at 21nt and 24nt. In total, 523 MIR-miRNA entries (considering both MIR gene origin and mature miRNA product) were identified (468 conserved and 55 novel), with conserved miRNAs from 48 miRNA families (Appendix A). The novel miRNAs were newly discovered miRNAs in durum wheat (where its reads were not registered in the miRBase, but could be mapped to the durum wheat genome and could form a secondary structure of a miRNA hairpin). 

The Venn diagrams (Figure 1) illustrate the distribution of miRNAs between different biological groups. When comparing across treatment groups at each time-point (Figure 1a,b), both genotypes appeared to have more miRNAs that were exclusively expressed in the CG (control group, see Table 1 for a summary of terms used in library names) at early stages of stress (5 DPA, 5 days post-anthesis). As stress continued to 25 DPA, for DBA Aurora (stress-tolerant genotype), the highest number of exclusively expressed miRNAs was found under WH (water-deficit plus heat stress), while for L6 (stress-sensitive genotype), it was found under HS (heat stress). Closer to maturity (45 DPA), the highest number of exclusively expressed miRNAs was found under WS (water-deficit stress) for L6, while for DBA Aurora, it was under HS. Interestingly, at all time-points (except for 45 DPA), DBA Aurora always had a lower number of miRNAs that were commonly expressed under all treatments compared with L6, suggesting that in DBA Aurora, more miRNAs showed expression specificity to different treatment groups. Similar results could be found when looking at the distribution of miRNAs across different time-points within treatments (Appendix A). In all groups, DBA Aurora also always had a lower number of miRNAs that were commonly expressed at all time-points compared with L6, suggesting that in DBA Aurora, more miRNAs showed expression specificity to different reproductive stages. 

### 2.2. DEMs Subject to Stress Type, Genotype, and Time-Point

To identify DEMs subject to stress type, miRNA expression was analyzed across treatment groups within each time-point. Heat maps were generated to show DEMs with a *p* < 0.01 at each time-point (Appendix A). In DBA Aurora (Appendix A), 160 miRNAs showed significant differential expression (*p* < 0.05) across four treatment groups at 5 DPA, 78 at 15 DPA, 154 at 25 DPA, 129 at 35 DPA, and 187 at 45 DPA. In L6 (Appendix A), 225 miRNAs showed significant differential expression (*p* < 0.05) across four treatment groups at 5 DPA, 155 at 15 DPA, 196 at 25 DPA, 154 at 35 DPA, and 213 at 45 DPA. 

To identify DEMs subject to genotype, miRNA expression was compared between two genotypes within each treatment at five time-points (Appendix A). Interestingly, there were always a higher number of DEMs that were less abundant in the stress-tolerant genotype (down-regulated) compared with the number of DEMs that were more abundant in the stress-tolerant genotype (up-regulated), irrespective of the *p*-value used (0.01 and 0.05) and the treatment condition (except for TL25_CG vs. SL25_CG). 

To identify DEMs subject to reproductive time-points, miRNA expression was compared across five time-points within treatments (Appendix A). In DBA Aurora, 236 miRNAs showed significant differential expression (*p* < 0.05) across five time-points in control, 237 under WS, 207 under HS, and 222 under WH. In L6, 201 miRNAs showed significant differential expression (*p* < 0.05) across five time-points in control, 249 under WS, 271 under HS, and 277 under WH.

### 2.3. Transcriptome Sequencing and the Identification of Stress-Responsive DEGs

Sequencing of eight transcriptome libraries generated over 649 million valid reads (Appendix A). A total of 49,436 genes and 117,637 transcript isoforms were identified (Appendix A). The median abundance of genes in the eight libraries ranged from 2.80 to 3.46 fragments per kilobase million (FPKM), and the mean abundance ranged from 17.07 to 19.32 FPKM. The median abundance of transcripts in the eight libraries ranged from 0.65 to 0.76 FPKM, while the mean abundance ranged from 8.98 to 10.40 FPKM (Appendix A). 

To identify stress-responsive DEGs, comparisons were made between the control group and each stress group within each genotype (Appendix A). In DBA Aurora, 229 genes were significantly regulated (*p* < 0.05) under WS, 5114 genes under HS, and 2510 genes under WH. In L6, 2010 genes were significantly regulated (*p* < 0.05) under WS, 8410 genes under HS, and 6351 genes under WH. These stress-responsive DEGs included a wide range of transcription factors, plant growth regulators, hormone signaling receptors, and key metabolic enzymes.

GO (Gene Ontology) enrichment analysis was performed for stress-responsive DEGs in the two genotypes, with several GO terms common to both, while some were specific to the stress-tolerant or -sensitive genotype (Figure 2). Under WS, GO terms, like gibberellic acid-mediated signaling pathway (GO:0009740) and regulation of chlorophyll biosynthetic process (GO:0010380), were only significantly enriched in DBA Aurora (Figure 2a). Terms like response to karrikin (GO:0080167) and protein serine/threonine kinase activity (GO:0004674) were enriched in L6. Interestingly, a few carbohydrate-related terms with a relatively high rich factor were L6-specific, including starch synthase activity (GO:0009011) and glycogen (starch) synthase activity (GO:0004373). Under HS, DBA Aurora and L6 shared photosynthesis-related GO terms with a high rich factor, such as chlorophyll-binding (GO:0016168) and photosynthetic electron transport in photosystem I (GO:0009773) (Figure 2b). GO terms, including response to karrikin (GO:0080167) and poly(U) RNA binding (GO:0008266), were only highly enriched in DBA Aurora under HS. Under WH, the two genotypes still shared some photosynthesis-related terms (Figure 2c), but DBA Aurora had specific terms like protein folding (GO:0006457) and response to hydrogen peroxide (GO:0042542). Interestingly, a few ribosome-related terms were only enriched in L6, such as structural constituent of ribosome (GO:0003735) and cytosolic large ribosomal subunit (GO:0022625).

Similarly, KEGG (Kyoto Encyclopedia of Genes and Genomes) pathway analysis of the stress-responsive DEGs suggested that some major pathways were shared, while others were genotype-specific (Appendix A). Under WS (Appendix A), DBA Aurora-specific pathways included plant hormone signal transduction (ko04075) and phenylpropanoid biosynthesis (ko00940), while L6-specific pathways included starch and sucrose metabolism (ko00500) and MAPK signaling pathway—plant (ko04016). Both genotypes had photosynthesis-related KEGG pathways under HS, such as carbon fixation in photosynthetic organisms (ko00710) (Appendix A). Pathways like glycerophospholipid metabolism (ko00564) and ABC transporters (ko02010) were specific to DBA Aurora, while the phosphatidylinositol signaling system (ko04070) and purine metabolism (ko00230) were L6-specific. Under WH, pathways activated in both genotypes included valine, leucine, and isoleucine degradation (ko00280), starch and sucrose metabolism (ko00500), and several photosynthesis-related pathways (Appendix A). DBA Aurora had specific pathways, including ABC transporters (ko02010), while L6 had specific pathways, such as cysteine and methionine metabolism (ko00270) and ribosome biogenesis in eukaryotes (ko03008). 

### 2.4. DEGs with Genotypic Expression

To identify genotypic DEGs, gene expression was compared between two genotypes within treatments. In total, 1307 genes showed differential expression between DBA Aurora and L6 under the CG, 990 genes under WS, 2714 genes under HS, and 3250 genes under WH (Appendix A). The volcano plots and bar chart show the DEGs that were up-regulated (more abundant in DBA Aurora) or down-regulated (less abundant in DBA Aurora) under each treatment (Appendix A). Interestingly, there were always a higher number of genes that were more abundant in DBA Aurora (up-regulated) than those that were less abundant (down-regulated), irrespective of treatment (Appendix A). Such a genotypic pattern of DEGs was the opposite pattern observed for DEMs (Appendix A) at 5 DPA. This was expected considering the possible miRNA-induced silencing effect on the mRNA population.

GO enriched analysis was performed for genotypic DEGs within treatments (Appendix A). In control (Appendix A), GO terms that were enriched included signal transduction (GO:0007165) and cell surface receptor signaling pathway (GO:0007166). Under WS (Appendix A), enriched GO terms included water channel activity (GO:0015250), cellular water homeostasis (GO:0009992), and gibberellin biosynthetic process (GO:0009686); while under HS (Appendix A), enriched terms were mostly chloroplast or photosynthesis-related. Under WH (Appendix A), apart from the chloroplast-related terms, other highly enriched terms included RNA methyltransferase activity (GO:0008173) and nuclear mRNA surveillance (GO:0071028). The results revealed the various biological pathways that were differentially activated between two genotypes in response to single and combined stress conditions. 

### 2.5. SNP (Single Nucleotide Polymorphisms) and AS (Alternative Splicing) Analysis 

In each transcriptome library, we detected 55,740 to 59,265 single nucleotide variants (SNVs), and 943 to 1068 INDELs in gene regions (Appendix A). Exonic SNVs and INDELs were further annotated for their effects. A higher number of synonymous SNVs (from 28,661 to 30,222) compared to non-synonymous SNVs (26,310 to 28,418) in each library was observed, and a higher number of stop-gain SNVs (390 to 448) than start-loss SNVs (152 to 192). For INDELs, a higher number of frameshift deletions (351 to 401) compared to frameshift insertions (135 to 165) in each library was noted, and a higher number of non-frameshift deletions (247 to 295) than non-frameshift insertions (175 to 219). 

A total of 219,140 AS events were identified, classified into 12 common AS types (Appendix A): exon skipping (SKIP), cassette exons (MSKIP), retention of single (IR) and multiple (MIR) introns, alternative exon ends (AE), alternative transcription start site (TSS), alternative transcription termination site (TTS), approximate SKIP (XSKIP), approximate MSKIP (XMKIP), approximate IR (XIR), approximate MIR (XMIR), and approximate AE (XAE). TSS and TTS had the highest number of AS events. Regardless of genotype, WS appeared to reduce the number of AS events (compared with CG), while HS resulted in the highest number of AS events among all groups, with WH reporting the second-highest number of AS events.

### 2.6. Degradome Signatures under Different Types of Stress

Around 506.80 million raw reads were obtained from degradome libraries, of which 228.71 million were mapped to input cDNA reference (Appendix A). In DBA Aurora, 102,490 target transcript sites were identified (Appendix A). In L6, 111,804 target transcript sites were identified (Appendix A). 

To analyze the miRNA-induced mRNA degradation pattern in response to stress, expression levels of mRNA degradome tags were analyzed (Appendix A). In DBA Aurora, 45,333 mRNA targets showed a differential degradation pattern in response to WS (19,909 up-regulated, 25,424 down-regulated; where up-regulation represents a higher degree of degradation induced by miRNAs, and down-regulation represents lower mRNA degradation); under HS, there were 49,953 targets (23,929 up-regulated, 26,024 down-regulated), while under WH, there were 47,992 targets (22,375 up-regulated, 25,617 down-regulated). In L6, 44,570 mRNA targets showed a differential degradation pattern in response to WS (18,495 up-regulated, 26,075 down-regulated), while under HS, there were 53,267 targets (26,119 up-regulated, 27,148 down-regulated). In the WH treatment, there were 56,642 targets (31,016 up-regulated, 25,626 down-regulated). The number of targets that showed a down-regulated degradation pattern was always higher than the number of targets with up-regulated patterns in both genotypes under three stress treatments (with the exception in L6 under WH). 

Enrichment scatter-plots showed the significant GO terms enriched for these targets with stress-responsive patterns (Appendix A). While several GO terms were shared between genotypes, others were genotype-specific. Under WS (Appendix A), both genotypes had enriched terms like leaf senescence (GO:0010150) and GTPase activity (GO:0003924). DBA Aurora-specific terms included cell redox homeostasis (GO:0045454) and response to water deprivation (GO:0009414). L6-specific terms included stomatal closure (GO:0090332) and the oxidation-reduction process (GO:0055114). Under HS (Appendix A), DBA Aurora-specific terms included a response to hydrogen peroxide (GO:0042542) and stomatal closure (GO:0090332). L6-specific terms included protein binding (GO:0005515) and leaf senescence (GO:0010150). Under WH (Appendix A), DBA Aurora-specific terms included the tricarboxylic acid cycle (GO:0006099) and NADP binding (GO:0050661). L6-specific terms included elongation factor activity (GO:0003746) and protein homodimerization activity (GO:0042803). Such results revealed the complexity of the biological processes modulated by miRNA-regulated mRNAs specific to different stresses in two genotypes. 

### 2.7. Multi-Omics Analysis: Stress-Responsive miRNA-mRNA Modules 

Key miRNA-mRNA modules with antagonistic regulatory patterns subject to different stresses were discovered based on the integrated multi-omics analysis [66,67,70,71]. miRNA-mRNA interactions of these modules were evidenced by three sequencing datasets (miRNA expression evidenced by sRNA-seq results, mRNA expression evidenced by transcriptome-seq results, and miRNA-guided mRNA cleavage evidenced by degradome-seq results). Co-expression pattern recognition of these key modules was based on statistically significant differential expression (*p* < 0.05) of miRNAs/mRNAs with an antagonistic regulatory pattern (i.e., significantly up-regulated miRNAs corresponding to significantly down-regulated mRNA targets under stress, or vice versa). 

In DBA Aurora (Appendix A), ten miRNA-mRNA pairs showed significant antagonistic regulatory patterns in response to WS, 900 pairs for HS, and 408 pairs for WH. In L6 (Appendix A), a total of 179 miRNA-mRNA pairs showed significant antagonistic regulatory patterns in response to WS, 1795 pairs for HS, and 1487 pairs for WH. Interestingly, there were always a higher number of miRNA-mRNA pairs, where miRNAs were down-regulated (correspondingly mRNAs were up-regulated), than the number of pairs, where miRNAs were up-regulated (correspondingly mRNAs were down-regulated), except for DBA Aurora under WS (where only limited pairs were identified). These results suggested that the activation of protein-coding genes (possible positive regulators of stress adaption) was favored over suppression of genes (possible negative regulators of stress adaptation). A significant number of stress-responsive mRNAs with antagonistic patterns were transcription factor (TF) families and included DREB, NAC, ARF, WRKY, MYB, MYC, bZIP, NF-YC, ERF, bHLH, PIF, GSK, GATA, HOX, DRF-like, and PCL family members (Appendix A). Among these miRNA-TF pairs, 113 were newly discovered miRNA-target pairs. In other crop plants and model plant species, TF families, including DREB, NAC, ARF, WRKY, MYB, bZIP, bHLH, received significant attention and functionally validated to hold significance in miRNA-conferred stress tolerance and plant development [31,72,73,74,75,76,77]; therefore, the related durum wheat miRNA-TF pairs and their regulatory patterns were summarized here (Table 2). Future research could focus on the functional validation and characterization of these miRNA-TF pairs and their downstream genes, which could serve as promising candidates for enhancing stress tolerance in tetraploid wheat. The T-plots for these miRNA-TF pairs showed the mRNA cleavage sites of TF genes silenced by miRNAs (Appendix A), validated by degradome sequencing. 

KEGG pathway classification of these miRNA-mRNA modules revealed the key biological processes regulated in response to each type of stress (Figure 3). Most miRNA-mRNA pairs were involved in metabolism-related KEGG pathways. Under WS (Figure 3a), the highest percentage of KEGG pathways activated in DBA Aurora was lipid metabolism; for L6, it was carbohydrate metabolism. Under HS (Figure 3b), both genotypes had a high percentage of pathways classified under carbohydrate metabolism, lipid metabolism, amino acid metabolism, and biosynthesis of other secondary metabolites. A similar pattern was identified under WH (Figure 3c). However, for DBA Aurora, more miRNA-mRNA pairs were involved in the biosynthesis of other secondary metabolites than lipid metabolism. For L6, a higher percentage was involved in translation compared with folding, sorting, and degradation under genetic information processing pathways.

### 2.8. Multi-Omics Analysis: Genotype-Dependent miRNA-mRNA Modules 

Genotype-dependent miRNA-mRNA pairs (DBA Aurora vs. L6) with antagonistic regulatory patterns were identified under each treatment (Appendix A). In the CG, 273 miRNA-mRNA pairs showed significant genotypic patterns. Among these, 198 pairs exhibited down-regulation of miRNAs (less abundant in DBA Aurora) and up-regulation of mRNAs targets (more abundant in DBA Aurora), while 75 pairs exhibited the opposite, where miRNAs were up-regulated, but the mRNAs targets were down-regulated. Under WS, 124 miRNA-mRNA pairs were genotype-dependent, with 74 pairs exhibiting down-regulation of miRNAs with up-regulation of mRNAs targets, and 50 pairs exhibited the opposite pattern. Under HS, 340 miRNA-mRNA pairs were genotype-dependent, with 225 pairs exhibiting down-regulation of miRNAs and up-regulation of mRNAs targets, while 115 pairs exhibited the opposite pattern. The highest number of genotypic miRNA-mRNA pairs (468) were found under WH, where 160 pairs exhibited down-regulation of miRNAs and up-regulation of mRNAs targets, and 308 pairs exhibited the opposite trend. Interestingly, under all conditions except for WH, there were always a higher number of miRNA-mRNA pairs where mRNA expression was higher in DBA Aurora due to the lower abundance of miRNA. Figure 4 displays the distribution of KEGG pathways among up-regulated and down-regulated miRNA-target pairs under different treatments. When heat stress was present (HS and WH treatments), mRNAs that were involved in the carbon fixation in photosynthetic organisms (ko00710) were more abundant in DBA Aurora (up-regulated), while target genes involved in protein processing in the endoplasmic reticulum (ko04141) were lower in DBA Aurora (down-regulated).

### 2.9. qPCR Analysis of DEMs and DEGs

Genotypic patterns were observed for some of the expression profiles of ten stress-responsive miRNAs and 15 targets (Figure 5). For example, osa-miR160f-5p and tae-MIR9772-p5 were both significantly down-regulated in DBA Aurora under all stresses but were significantly up-regulated in L6 under HS and WH. Stress-responsive miRNA and their targets sometimes exhibited antagonistic patterns, subject to genotype and stress type. For example, in DBA Aurora, tae-MIR9662b-p5_1ss9CG was down-regulated in response to all stress treatments, and its target transcription factor bHLH47 was up-regulated correspondingly under all stress conditions. Consistent with previous studies [45,78], miRNAs and their targets did not always exhibit a negative correlation in their expression. As the regulatory relationships of miRNA-target modules were not always one-on-one specific, one miRNA could regulate multiple mRNAs, and one mRNA could be synergistically targeted by several miRNAs. 

## 3. Discussion

### 3.1. miRNA Expression Specificity Provides New Insights into Their Regulatory Roles Specific to Genotype and Developmental Stage

Although miRNAs have been extensively studied in many cereal crops, no research has been performed to compare miRNA expression against multiple factors, including genotypes, different stress types, and multiple time-points during reproduction. In this study, we discovered high expression specificity of different miRNAs to all the above-mentioned factors. Such patterns suggest that miRNAs not only play a key part in genotype by environment interactions but also in cereal reproductive development, as well as providing possible feedback to small RNA biogenesis pathways. Among all the miRNAs identified (Appendix A), 24 miRNAs were exclusively expressed in DBA Aurora, and 59 miRNAs were only expressed in L6. Further investigation of the function of these specific miRNA targets provided valuable information on the regulatory pathways available for any given genotype. For example, ata-miR167c-3p was only expressed in DBA Aurora under HS (Appendix A). One target of ata-miR167c-3p is a stomatal closure-related actin-binding protein 1 (SCAB1, TRITD_3Bv1G186570), which provides precise regulation of actin filaments and stomatal movement [79]. The regulation of *SCAB1* expression by ata-miR167c-3p could potentially contribute to adaptive and coordinated stomatal closure under HS in DBA Aurora, which would not be available for the stress-sensitive genotype L6. Indeed, under stress, stomatal conductance reduced to a greater extent in L6 when compared with DBA Aurora, where stomatal conductance was more tightly controlled [8].

For developmental stage-specific miRNAs, 11 miRNAs were exclusively expressed in 5 DPA libraries, seven in 15 DPA libraries, 28 in 35 DPA libraries, and 16 in 45 DPA libraries. These time-point specific miRNAs have possible functional roles in regulating biological processes specific to different reproductive stages from flowering to grain maturity. For example, tae-miR9666a-3p was only expressed at 5 DPA (Appendix A). It targets a *PLATZ* (*plant AT-rich sequence and zinc-binding protein)* transcription factor gene (TRITD_7Bv1G163890). In rice, the PLATZ transcription factor positively regulates grain length and negatively regulates grain number formation by coordinating cell proliferation in young panicles and grains during early grain development [80]. The regulation of the PLATZ protein by tae-miR9666a-3p was only present 5 days after flowering, suggesting that the miRNA-mRNA pair could serve a similar function in durum wheat. Future research could strategically examine the spatial-temporal expression of the tae-miR9666a-3p-PLATZ module in various tissues at different reproductive and grain development stages. 

### 3.2. Stress-Responsive DEMs and DEGs Reveal Synergistic Interactions between Water-Deficit Stress and Heat Stress Response

Stress-responsive patterns of conserved miRNA families have been described in many cereal species, as well as summarized in several reviews [23,29,30,81]. However, information of the same miRNA in response to various stress types was often collected from different studies. Here, we systematically described miRNA profiles in the same sample set subjected to single and combined water-deficit and heat stress. At certain time-points, some miRNAs exhibited the same responsive pattern to all stress treatments (down-regulated under all stresses or up-regulated under all stresses). For example, in DBA Aurora, ata-miR528-5p was significantly down-regulated under all stresses at 5 DPA and 45 DPA (Appendix A). miR528 targets several copper-containing oxidases that contribute to cellular redox homeostasis, such as Cu–Zn SOD (superoxide dismutase). Thus, repressing miR528 expression to promote SOD activity could be a common response to different stresses, in line with previous findings in durum wheat and bread wheat [41,82,83]. Some miRNAs exhibited contrasting regulatory patterns towards three stress treatments. For example, another miRNA that targets SODs, osa-miR398a, was up-regulated under WS and HS but was down-regulated under WH in DBA Aurora at 15 DPA (Appendix A). It was also up-regulated under HS at 25 DPA in DBA Aurora but was down-regulated under WS and WH. Our results further confirmed that the impacts of combined stress on the miRNAome could not be simply extrapolated based on their impacts taken individually, and it is subject to not only the developmental stages of the plant but also the genotype, even within the same species. To further elucidate the synergistic interactions of water-deficit and heat stress response at the transcriptional level, the expression profiles of miRNAs should be analyzed together with the mRNA transcriptome profile. 

Similar to miRNAs, mRNA changes in response to combined stress were orchestrated and sometimes unique. Firstly, the numbers of DEGs identified were quite different under three stress treatments. HS triggered the highest number of DEGs in both genotypes, and the lowest number of DEGs were found under WS, with the DEG number under the combined stress being in between these values. This could be explained by the fact that the heat stress applied was an intense shock (37 °C/27 °C compared with the control at 22 °C/12 °C) and transient in nature (24 h). In contrast, the WS applied was mild water-limiting stress, where soil water content was kept at 50% of the field capacity (rather than drought stress, which is usually at <30% of field capacity). The WS treatment was also chronic, where it started at the booting stage (approximately 20 days before flowering). Acute short-term stress tended to induce higher transient gene expression changes than chronic stress exposure, where plants had time for acclimation with possible implications from the stress-priming [84,85]. Indeed, HS caused more damage to the plants than WS, as was evidenced by significantly lower yield components (such as biomass, grain weight, and fertility) recorded in DBA Aurora [8]. 

The effect of combined stress on DEGs was not simply additive from single stress conditions. Similar to DEM profiles, some DEGs were commonly up- or down-regulated under all stresses, while others exhibited unique patterns specific to either single or combined stress. The commonly regulated genes include many well-known families often induced by heat and water-deficiency, such as heat shock protein genes, dehydrins, and key metabolic enzymes. As an example, a *protein phosphatase 2C* (*PP2C*) gene (TRITD_5Av1G133330) was significantly up-regulated under all stresses in DBA Aurora (Appendix A). PP2Cs play critical roles in signaling by abscisic acid (ABA), a hormone well known to coordinate developmental signals and environmental cues, such as drought and heat [86]. In Arabidopsis, overexpression of a rice *PP2C* gene enhanced ABA insensitivity and tolerance to salt, mannitol, and drought stress [87]. In the current study, the common regulatory patterns of *PP2C* in DBA Aurora (up-regulated under all stresses) suggested that PP2C-coordinated ABA signaling pathways could be in play under all conditions, contributing to stress adaptation to not only single but also combined stress. For genes that were specifically regulated under combined stress, one example is a *Bowman-Birk type trypsin inhibitor* gene (*BBI*, TRITD_1Av1G004660). This gene only exhibited a significant up-regulation pattern under WH in DBA Aurora (Appendix A). A very recent study in Arabidopsis demonstrated that BBIs play important roles in stress adaptation via alleviating cellular oxidative stress [88]. Transgenic lines overexpressing the maize *BBI* gene showed significantly stronger trypsin-inhibitor activity, leading to enhanced antioxidant enzyme activities that contributed to the maintenance of higher leaf relative water content (RWC) under drought stress. Indeed, in our previous report [8], among all the Australian genotypes studied, DBA Aurora was the only genotype that managed to maintain a high leaf RWC under WH at 5 DPA (with no significant difference to the control). Future research can extend this work by investigating BBI expression in various varieties and elite breeding lines, along with their serine protease inhibitory activity, antioxidant enzyme levels, and leaf water status traits under various stresses. 

KEGG pathway and GO analysis of DEGs provided a thorough picture of the common and unique pathways activated under WS, HS, and WH. Several photosynthesis-related GO terms were enriched under all stresses, such as photosystem I (GO:0009522) and II (GO:0009523) and chlorophyll-binding (GO:0016168). The results were consistent with findings in bread wheat that key genes in photosynthetic processes were most commonly affected by individual and combined drought and heat stress [57]; and that reproductive stage drought and heat stress severely impacted photosynthesis and transpiration activities, leading to a significant loss in grain number and grain size [8,89]. In Australian durum wheat genotypes [8], interactions of the two stresses combined were often pronounced, with impacts on yield loss often higher than individual stress, but usually lower than their additive sum. However, stress-tolerant genotypes like DBA Aurora still managed to maintain certain physiological and yield traits (such as chlorophyll content and harvest index) under combined stress, with no significant difference to its performance under single stress conditions. Such yield stability more than likely depends on the genotypic specificity of photosynthetic gene expression upon stress.

Certain regulatory pathways were only shared between one single stress and the combined stress. For example, many GO terms common to HS and WH were related to cellular protein protection, such as protein refolding (GO:0042026) and cellular response to unfolded proteins (GO:0034620). Terms shared between WS and WH were more related to the maintenance of cellular water status, such as cellular water homeostasis (GO:0009992) and water channel activity (GO:0015250). In agreement with previous research, the adaptation process to heat and water-deficit stress in cereals could be quite divergent, with only some overlap in the regulatory pathways that govern these mechanisms [57,90]. In barley, analysis of the leaf proteome revealed that drought mainly resulted in a specific increase of proteins with functions like cell detoxification and water homeostasis maintenance, while heat induced the production of heat shock proteins (HSPs) and other chaperon proteins to maintain correct protein configuration [90]. Indeed, our transcriptome profiling discovered a large number of HSPs (including chloroplastic or mitochondrial members) that were significantly up-regulated under HS and WH (Appendix A). Looking forward, future studies could investigate the production partitioning of proteins under combined stress comparing with single stress in a broad range of germplasm, alongside their gene expression profiles. This could be useful in elucidating how genotypes of varying tolerance employ different strategies to coordinate and partition common and unique biological pathways required for water-deficit and/or heat stress adaptation.

### 3.3. Stress-Responsive miRNA-mRNA Modules Are Valuable Attributes for Building Stress Tolerance

Gene regulation via sequence-specific recognition of miRNA-mRNAs offers a highly efficient mechanism for crops to cope with stress. The key stress-responsive miRNA-mRNA modules with antagonistic regulatory patterns subject to different stresses were verified by integrated co-expression analysis obtained from the three omics sequencing datasets, as previously described [66,67]. Here, we constructed a schematic view of key miRNA-RNA modules involved in water-deficit and heat stress response based on their molecular functions (Figure 6). The stress response started with signal perception, involving key components like the Ca2+ influx (Figure 6a). Examples of miRNA-mRNA modules activated at this stage included miRNAs targeting calcium-dependent protein kinases (CDPKs) and receptor-like protein kinases (RLKs). Thereafter, stress signal transduction was mediated by a number of signaling regulators, such as various phytohormones, including ABA and ethylene. Many durum wheat miRNAs were involved at this stage, via their regulation of hormone regulators like auxin response factors (ARFs), PP2Cs, and CBL (Casitas B-lineage Lymphoma)-interacting protein kinases (CIPKs). Downstream transcriptional changes involved a great number of transcriptional factors, which could provide feedback to fine-tune ongoing signal transduction. An extensive network of miRNA-TF pairs was identified at this stage, contributing to the stress signaling processes. From there, many positive stress regulators were activated, such as reactive oxygen species (ROS) scavenging enzymes, chaperone proteins, and protein phosphatases, to either alleviate cellular damage or adjust biological processes, leading to adaptive physiological changes. The miRNA-guided stress-responsive networks were multi-layered, where one miRNA could regulate multiple mRNAs, and one mRNA could be synergistically targeted by several miRNAs. Figure 6b illustrates the multiple-to-multiple regulatory connections between miRNAs and their target genes using Cytoscape, as previously described [67,91,92]. Such miRNA-mRNA connections existed in both genotypes; however, the expression pattern (down-regulated or up-regulated) of these miRNAs/genes could be different (or even opposite) between genotypes in response to stress, which could contribute to the difference in their tolerance levels. Feedback loops fine-tuned the homeostasis of miRNA and target levels together with other regulating factors; thus, miRNA/mRNA expression could vary under different conditions between genotypes to achieve diverse biological interactions [23,30]. Such connections, shown in Figure 6b, also provide useful information when considering the genetic manipulation of a specific miRNA, to identify other connected components that could be synergistically affected. 

Some miRNA-mRNA modules with key functions exhibited contrasting patterns between stress-tolerant and -sensitive durum wheat genotypes, suggesting that they could be valuable attributes for building stress-tolerant germplasm. As an example, qPCR showed that osa-miR160f-5p was significantly down-regulated in DBA Aurora under all stresses, but was up-regulated in L6 under HS and WH (Figure 5a). Similarly, ata-miR396c-5p was significantly down-regulated in DBA Aurora under WS but was up-regulated in L6 under WS and WH. Correspondingly, one target of ata-miR396c-5p, an NADH dehydrogenase (NDH), was significantly up-regulated under HS and WH in DBA Aurora but was down-regulated in L6 under WS and HS (Figure 5b). A newly discovered target of osa-miR160f-5p, chloroplastic PGR5-like protein, was significantly up-regulated under HS and WH in both genotypes, but with a higher fold-change in DBA Aurora under WH. NDH and PGR5 are key components in the photosynthetic energy conversion process. They each mediate two major pathways of the cyclic electron flow, playing significant roles in photosynthetic electron-transport pathways [93]. The increase of NDH and PGR5 under abiotic stress has been observed in a few flowering species [94,95,96]. NDH and PGR5 activities both increased greatly (especially PGR5) in response to drought, heat, and high illumination. The stimulation of PGR5-dependent and NDH-dependent cyclic pathways helped to protect the photosynthetic apparatus against stress, resulting in lower levels of ROS accumulation in the leaves [94]. In durum wheat, the increase of *NDH* and *PGR5* expression via a reduced abundance of osa-miR160f-5p and ata-miR396c-5p could serve a similar purpose to protect the photosynthetic apparatus, as shown by high chlorophyll content observed in DBA Aurora under stress [8]. 

The conserved ata-miR396c-5p could contribute to plant fitness through its targets ClpC2 and HSP90 upon heat exposure. The chloroplastic chaperone protein gene *ClpC2* was significantly up-regulated under WH in DBA Aurora (Figure 5b). The *HSP90* gene was significantly up-regulated under HS in DBA Aurora, and under HS and WH in L6. The Clp/HSP100 chaperones and HSP90s are key components in cellular protein quality control. Specifically, functions of chloroplastic ClpC chaperones include protein import, support of downstream quality control, and chloroplast biogenesis [97]. HSP90s help other native proteins to maintain the correct folding structure and to prevent denaturation damage caused by abrupt or gradual temperature increase [15,98]. Accumulation of HSPs is a universal adaptive response to heat stress in many crops. Evidence has confirmed that overexpression of HSP90 can confer abiotic stress tolerance [99,100]. Therefore, the accumulation of these chaperon proteins via stress-reduced miRNA levels could provide an alternative for improving stress tolerance in breeding without the transgenic approach. 

Under stress, the ata-miR528-5p-target module could contribute to redox homeostasis with genotypic preference. ata-miR528-5p was down-regulated in DBA Aurora under HS and WH, with no significant change in L6 (Figure 5a). In accordance, its targets—an F-box protein and a Cu–Zn SOD—were both significantly up-regulated under WH in DBA Aurora (Figure 5b). As core components of the Skp1-Cullin-F-box (SCF) E3 ligase complex, F-box proteins are key elements in abiotic stress responses. In bread wheat, overexpression of the F-box gene *TaFBA1* led to enhanced tolerance to drought, heat, and oxidative stress [101,102,103]. Plants with high F-box levels exhibited higher relative water content, less chlorophyll degradation, better photosynthetic capacity, and less growth inhibition. Notably, TaFBA1 transgenic plants also had higher SOD activity, which enhanced the ROS scavenging ability. The positive impacts on cellular redox via the miR528-mediated F-box protein and SOD activity were specific to the stress-tolerant genotype, but in another case, the regulation of ROS scavenging by osa-miR398a appeared to be a common response. osa-miR398a was significantly down-regulated under WS and HS in DBA Aurora, and under HS in L6 (Figure 5a). Its target, a chloroplast copper chaperone for SOD, was significantly up-regulated in both genotypes under all stress conditions (Figure 5b). Copper chaperone proteins are essential for the delivery and the activation of Cu–Zn SOD [104,105]. The positive roles of miR528 and miR398 modules in maintaining redox homeostasis make them promising candidates for improving stress tolerance in durum wheat.

A newly-discovered miRNA-ARF module could also contribute to stress adaptation in durum wheat. Various modules of miRNA-directed regulation of ARFs have been reported before. For example, miR160 targets ARF8, ARF10, ARF16, ARF17, and ARF18; miR167 targets ARF6, ARF12, ARF17, and ARF25 [37,78,106]. Here, we discovered a new pairing of tae-miR408_L-1 targeting ARF17. tae-miR408_L-1 was significantly down-regulated in DBA Aurora under all stresses, with no significant change in L6 (Figure 5a). The *ARF17* gene was significantly up-regulated under WH in both genotypes, with a higher fold-change observed in DBA Aurora (Figure 5b). ARFs are critical components of auxin signaling, which has important roles in modulating various biological processes, such as lateral root initiation, shoot elongation, embryo patterning, and reproductive fertility [107]. ARFs bind to the conserved auxin response element in the promoters of auxin-targeted genes to regulate their expression. Generally, the structure of the transcriptional regulatory region of each ARF member determines whether it is a transcriptional activator or repressor. Research in Arabidopsis showed that ARF17 is an essential transcriptional activator for reproductive development [108]. In rice, repressed expression of ARF17 and other auxin receptors under drought and heat stress was the cause of spikelet sterility and yield reduction [107]. The increase of *ARF17* expression via reduced miR408 level could be contributing to better reproductive development, as observed in DBA Aurora, where yield traits like fertility showed no significant difference between WH and CG groups [8]. Future research should investigate the expression changes of miR408-ARF17 level in somatic and reproductive tissues under different stresses, along with correlation analysis of reproductive traits, such as fertility, spikelet number, and grain number.

## 4. Materials and Methods 

### 4.1. Plant Materials, Stress Treatment, and Sample Collection

Stress-tolerant genotype DBA Aurora and stress-sensitive genotype L6 (University of Adelaide breeding line, UAD1301020-8) were grown in controlled glasshouse conditions, as previously described [8]. Briefly, the standard growing conditions were 22 °C/12 °C (day/night) with a 12 h photoperiod. Four treatment groups were control group (CG), pre-anthesis water-deficit stress group (WS), post-anthesis heat stress group (HS), and pre-anthesis water-deficit plus post-anthesis heat stress group (WH). All plants were well-watered from germination to booting. Water-deficit stress was applied to WS and WH by maintaining the soil water content at 6% (half of the field capacity) from booting until harvest. Heat stress was applied by placing plants in a growth chamber at 37 °C/27 °C with a 12 h photoperiod. HS and WH plants were treated with heat for 24 h at multiple reproductive time-points consecutively (5, 15, 25, 35, and 45 days post-anthesis (DPA)). After 24 h, plants were moved back to the standard conditions and remained there between each treatment time-point. Flag leaf samples were collected at each DPA. Total RNA was extracted from a total of 120 samples (2 genotypes × 4 treatments × 5 time-points × 3 biological replicates) using the Tri reagent (Sigma-Aldrich, North Ryde, Australia) and treated with TURBO DNase (ThermoFisher Scientific, Scoresby, Australia). RNA concentration, quality, and integrity were assessed by NanoDrop, gel electrophoresis, and Bioanalyzer.

### 4.2. Small RNA Sequencing and Data Analysis

The 120 sRNA libraries were constructed using the NEBNext^®^ Multiplex Small RNA Library Prep Kit, as previously described [46]. Sequencing was performed on Illumina NovaSeq 6000 by the Australian Genome Research Facility (Melbourne, Australia). The sample information of each library is listed (Appendix A). sRNA-seq, transcriptome-seq, and degradome-seq datasets were submitted to NCBI GEO database (accession number GSE152973). Conserved and novel durum wheat miRNAs were identified using the ACGT101-miR program (LC Sciences, Houston, TX, USA), as previously described [109]. For details, see Appendix A. All identified miRNAs were categorized into five groups (G1-5), with G1-4 being conserved miRNAs, and G5 being novel miRNAs. DEMs were identified based on the normalized reads count [110], subject to genotype, treatment, and time-point. ANOVA and t-test were used to identify DEMs with statistical significance (*p* < 0.05). For fold-change calculations, normalized reads count at 0 was reset as 0.001 where necessary. 

### 4.3. Transcriptome Sequencing and Data Analysis

High-quality RNA (RNA Integrity Number > 9) from the 5 DPA samples were used for transcriptome-seq and degradome-seq. Eight transcriptome libraries (Appendix A) were constructed using the Illumina mRNA-Seq sample preparation kit, as previously described [111]. Transcriptome libraries were sequenced on Illumina NovaSeq 6000 at LC-Bio (Hangzhou, China). Transcriptome data was processed, as previously described [111]. For details, see Methods S1. Normalized relative abundance was expressed in FPKM. DEGs were identified with the edgeR package with a *p*-value < 0.05. For fold-change calculations, the FPKM at 0 was reset as 0.001 where necessary. SNP (single nucleotide polymorphism) calling was performed using the mpileup function in SAMtools (http://samtools.sourceforge.net). Annovar (http://annovar.openbioinformatics.org) was used for SNP annotation. Alternative splicing events were classified into 12 types with the ASprofile software (https://ccb.jhu.edu/software/ASprofile/). The number of each AS event was calculated for each library.

### 4.4. Degradome Sequencing and Data Analysis

Eight degradome libraries (Appendix A) were constructed, as previously described [111]. Sequencing was performed on an Illumina HiSeq2500 at LC-Bio (Hangzhou, China). Degradome data was processed, as previously described [112]. The CleaveLand package V4.0 [113,114] and the ACGT101-DEG program (LC Sciences, Houston, TX, USA) were used to identify candidate targets of durum wheat miRNAs. For details, see Methods S1. All targets were grouped into five categories (category 0-4) based on the signature abundance at each transcript position [71,112]. The categories indicated the confidence level of target prediction, with category 0 having maximum confidence, and category 4 having minimum confidence. Reads abundance were normalized to transcripts per billion (TPB). 

### 4.5. Gene Ontology Enrichment, KEGG Pathway Analysis, and Multi-Omics Analysis

GO terms of DEGs (from transcriptome-seq) and mRNA targets (from degradome-seq) were annotated according to their biological process, cellular component, and molecular function. KEGG pathway analysis was performed to identify the biochemical pathways that DEGs and target genes were associated with. The statistical enrichment of GO terms and KEGG pathways was performed, as previously described [69]. The multi-omics analysis was based on the statistical recognition combining sRNA-seq, transcriptome-seq, and degradome-seq datasets. Data of the three omics were first integrated to identify the mRNA-target pairs that could be evidenced by the three sequencing datasets at the same time (i.e., miRNAs supported by valid sRNA-seq reads, mRNAs supported by valid transcriptome-seq reads, and miRNA-induced mRNA degradation supported by valid degradome-seq reads). After all the validated miRNA-target pairs were found, those where both miRNAs and mRNA targets exhibited significant differential patterns in expression analysis (either subject to stress type or with genotype) were identified (*p* < 0.05). To validate the miRNA silencing effect among the significant miRNA-target pairs, those miRNAs and mRNAs that exhibited antagonistic regulatory patterns were identified (i.e., up-regulation of miRNA corresponding to down-regulation of mRNAs, or down-regulation of miRNAs corresponding to up-regulation of mRNAs), subject to comparisons made against the treatment factor or the genotype factor. A network showing the multiple-to-multiple connections between miRNAs and their target genes was constructed using Cytoscape, as previously described [58,75,76].

### 4.6. qPCR Analysis of Stress-Responsive miRNAs and Targets

Ten stress-responsive miRNAs and 15 target genes were selected for qPCR analysis. cDNA was synthesized using the MystiCq microRNA cDNA Synthesis Mix Kit (Sigma-Aldrich), as previously described [18,78]. qPCR analysis was performed using the PowerUp SYBR Green Master Mix (ThermoFisher Scientific) on a ViiA7 Real-Time PCR machine using the 2^−ΔΔCT^ method. Durum wheat GAPDH gene was used as the reference gene, and three replicates were conducted. Each expression profile was calculated as the log2 value of the fold-change (abundance under stress/abundance under control). Data were represented as mean ± SE (three replicates). Statistical significance of *p* < 0.05 (*) and *p* < 0.01 (**) was determined. Primers are listed in Appendix A. 

## 5. Conclusions

In conclusion, the current study provides the first multi-omics NGS analysis of durum wheat miRNAs and mRNAs regulated at the post-transcriptional and transcriptional level in response to water-deficit and heat stress. Durum wheat miRNA-mRNA modules have shown distinct genotype, stress, and developmental stage-dependent patterns. The multi-layered stress-responsive networks mediated by miRNAs are no doubt highly complex but intricately coordinated. New findings provided by this research now lay the foundations for improving stress resilience in cereal crops via knowledge-driven epi-breeding.

## Figures and Tables

**Figure 1 ijms-21-06017-f001:**
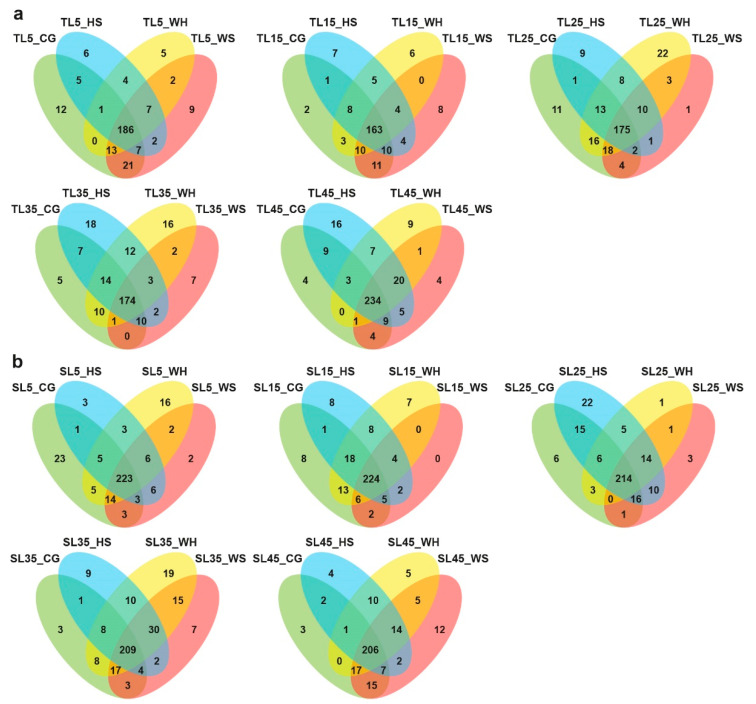
Venn diagrams of durum wheat microRNAs (miRNAs) identified between different biological groups. (**a**) miRNAs specific to treatment groups at each time-point in DBA Aurora. (**b**) miRNAs specific to treatment groups at each time-point in L6. TL, libraries made from leaf tissue of the stress-tolerant genotype DBA Aurora. SL, libraries made from leaf tissue of the stress-sensitive genotype L6. 5, 15, 25, 35, 45 indicate treatment time-points (days post-anthesis). CG, control group. WS, water-deficit stress group. HS, heat stress group. WH, water-deficit plus heat stress group.

**Figure 2 ijms-21-06017-f002:**
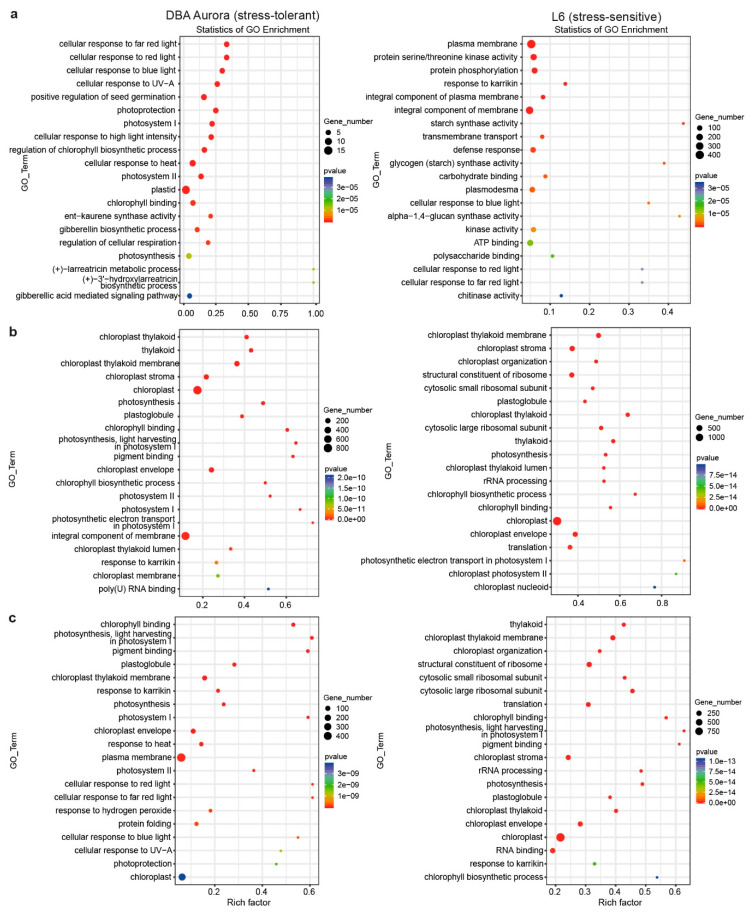
Gene ontology (GO) enrichment analysis of stress-responsive DEGs (differentially expressed genes, *p* < 0.05) under stress. (**a**) In response to water-deficit stress. (**b**) In response to heat stress. (**c**) In response to water-deficit plus heat stress. GO terms on the Y-axis are ranked by *p*-value in the enrichment analysis. Rich factor stands for the ratio of DEG number to the total number of genes that have been annotated under this GO term (i.e., the degree of enrichment).

**Figure 3 ijms-21-06017-f003:**
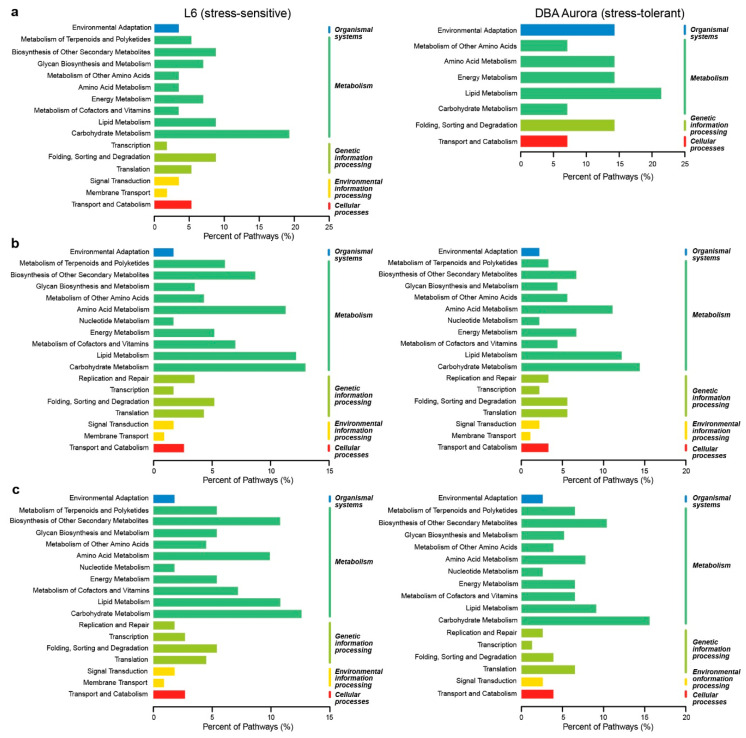
KEGG (Kyoto Encyclopedia of Genes and Genomes) pathway classification of stress-responsive microRNA-mRNA regulatory modules in two durum wheat genotypes. (**a**) In response to water-deficit stress. (**b**) In response to heat stress. (**c**) In response to water-deficit plus heat stress.

**Figure 4 ijms-21-06017-f004:**
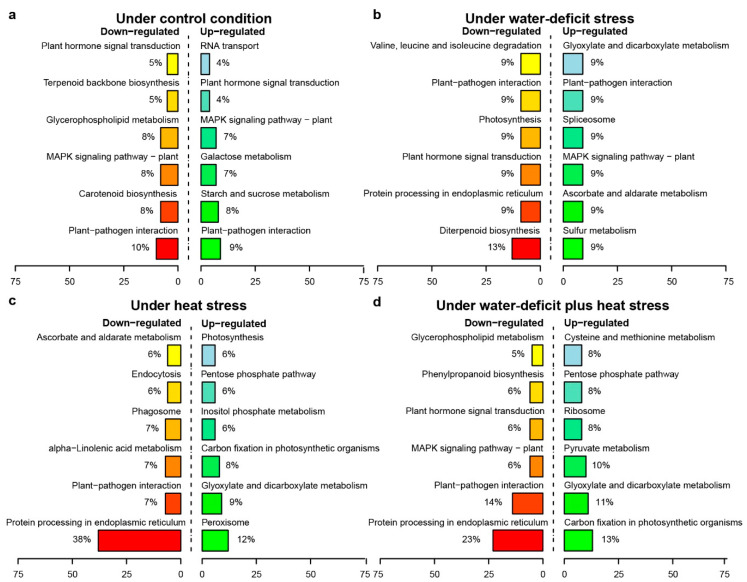
KEGG pathway classification of genotype-dependent miRNA-miRNA regulatory modules. (**a**) KEGG pathway distribution of miRNA-miRNA modules under the control condition. (**b**) KEGG pathway distribution of miRNA-miRNA modules under water-deficit stress. (**c**) KEGG pathway distribution of miRNA-miRNA modules under heat stress. (**d**) KEGG pathway distribution of miRNA-miRNA modules under water-deficit plus heat stress. Up-regulated, mRNA targets were more abundant in the stress-tolerant genotype DBA Aurora. Down-regulated, mRNA targets were less abundant in the stress-tolerant genotype DBA Aurora.

**Figure 5 ijms-21-06017-f005:**
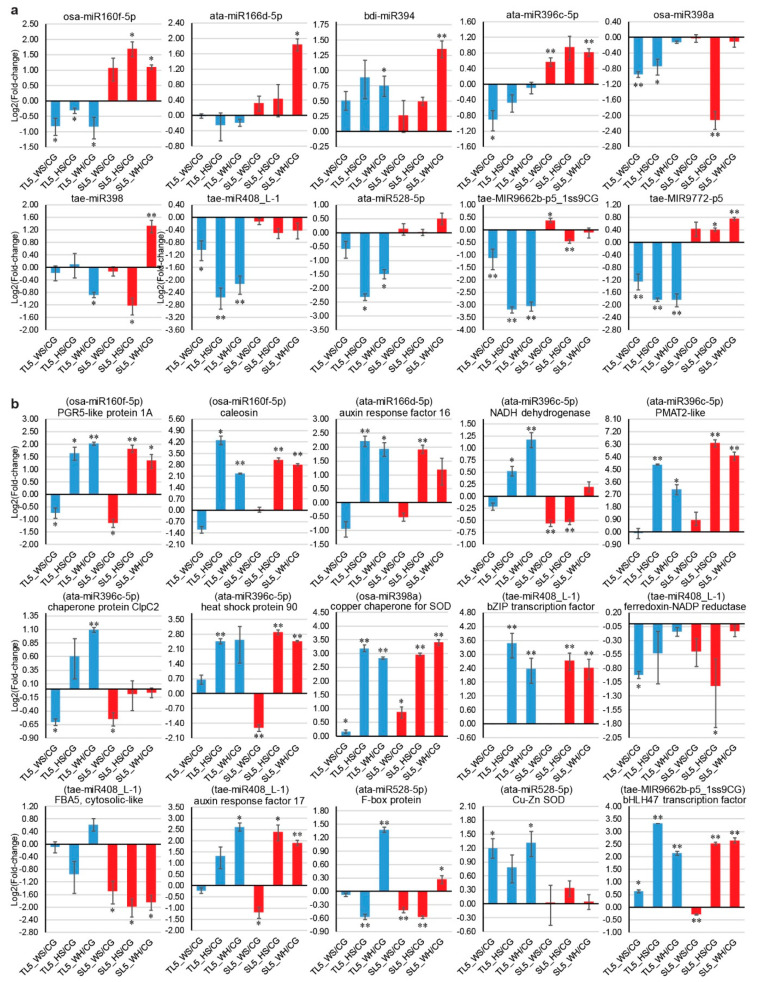
qPCR analysis of ten stress-responsive miRNAs and 15 target genes at 5 DPA (days post-anthesis). (**a**) Expression profiles of miRNAs. (**b**) Expression profiles of target genes. Each expression profile is shown as the log2 value of the fold-change (abundance under stress/control). Data are represented as mean ± SE (three replicates). * represents statistical significance of *p* < 0.05, ** represents statistical significance of *p* < 0.01. PGR5, proton gradient regulation 5. PMAT2, phenolic glucoside malonyltransferase 2. SOD, superoxide dismutase. FBA5, fructose-bisphosphate aldolase 5.

**Figure 6 ijms-21-06017-f006:**
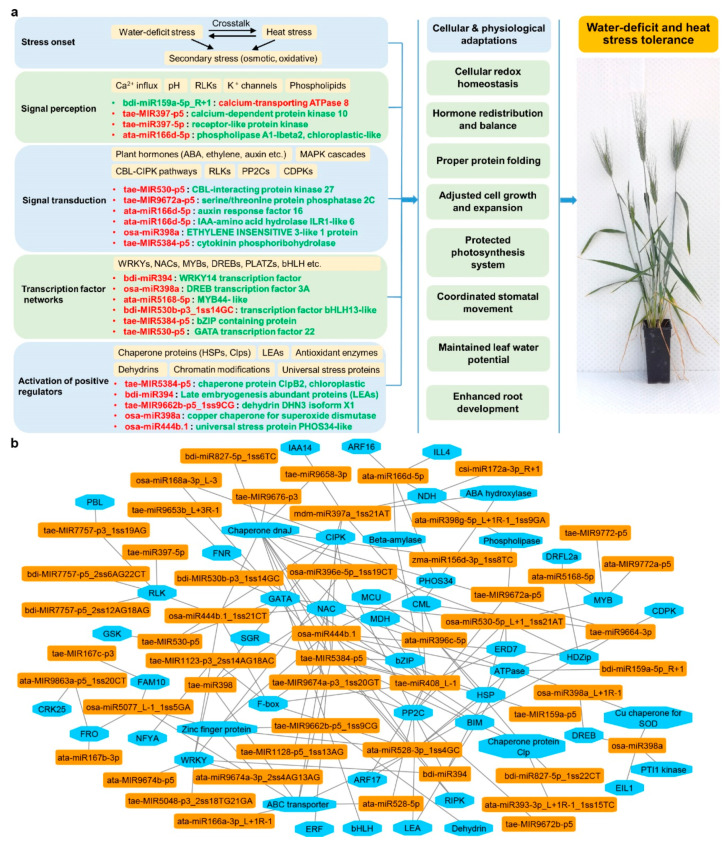
Water-deficit and heat stress response network mediated by key miRNA-RNA modules in durum wheat. (**a**) Examples of key miRNA-mRNA modules involved in the stress response networks. miRNA or gene names highlighted in green represent up-regulation under water-deficit plus heat stress; miRNA or gene names highlighted in red represents up-regulation under water-deficit plus heat stress; (**b**) multiple-to-multiple regulatory connections between miRNAs (orange) and their targets (blue).

**Table 1 ijms-21-06017-t001:** Summary of abbreviations used for next-generation library names.

Abbreviations	Description
CG	Control group
WS	Pre-anthesis water-deficit group
HS	Post-anthesis heat stress group
WH	Pre-anthesis water-deficit combined with post-anthesis heat stress group
TL	Libraries made from leaf tissue of the stress-tolerant genotype DBA Aurora
SL	Libraries made from leaf tissue of the stress-sensitive genotype L6
5, 15, 25, 35, 45	Time-points of sampling (DPA, days post-anthesis).

**Table 2 ijms-21-06017-t002:** Examples of stress-responsive microRNA (miRNA)-TF (transcription factor) pairs with antagonistic regulatory patterns in DBA Aurora. miRNA log2(FC) shows the log2(fold-change) value of the miRNA expression under stress vs. control. Target log2(FC) shows the log2(fold-change) value of the mRNA expression under stress vs. control. Pairing type indicates whether the miRNA-mRNA is newly discovered or conserved.

miRNA	Target Accession	Target Annotation	miRNA log2 (FC)	Target log2 (FC)	Stress Type	Pairing Type	TFs
osa-miR398a	TRITD_1Bv1G135100-4	DREB ^1^ transcription factor 3A	−1.23	2.95	WS	Newly discovered	DREB
			−2.67	1.75	HS	Newly discovered	
			−1.34	1.75	WH	Newly discovered	
mdm-miR397a_1ss21AT	TRITD_2Bv1G034790	NAC ^1^ transcription factor (plastid)	−inf ^2^	3.74	HS	Newly discovered	NAC
osa-miR398a_L+1R-1	TRITD_3Av1G235010-2	NAC transcription factor 6A	−2.05	4.13	HS	Newly discovered	
			−0.92	2.58	WH	Newly discovered	
osa-miR444b.1	TRITD_2Av1G026820	NAC transcription factor (plastid)	−1.77	3.40	HS	Newly discovered	
			−1.50	1.94	WH	Newly discovered	
	TRITD_5Av1G170450-2	NAC domain-containing protein 41	−1.50	2.90	WH	Newly discovered	
	TRITD_5Bv1G161590	NAC domain-containing protein 41	−1.50	3.59	WH	Newly discovered	
tae-MIR1128-p5_1ss13AG	TRITD_4Bv1G108860	NAC domain-containing protein 92-like	−inf	4.16	HS	Newly discovered	
tae-MIR5384-p5	TRITD_5Bv1G161590	NAC domain-containing protein 41	−1.98	3.59	WH	Newly discovered	
ata-miR166d-5p	TRITD_7Av1G051530-5	auxin response factor 16	−2.37	1.60	WH	Newly discovered	ARF ^1^
osa-miR167a-5p_R+1	TRITD_6Av1G046710	auxin response factor 6-like	−1.00	1.59	HS	Conserved	
tae-miR408_L-1	TRITD_7Bv1G194180-3	auxin response factor 17	−3.62	2.21	WH	Newly discovered	
ata-miR528-3p_1ss4GC	TRITD_7Bv1G194180-3	auxin response factor 17	−inf	1.81	HS	Newly discovered	
			−2.60	2.21	WH	Newly discovered	
	TRITD_7Av1G245930-4	auxin response factor 17	−2.60	1.79	WH	Newly discovered	
bdi-miR394	TRITD_1Bv1G030900-5	WRKY14 ^1^ transcription factor	−0.76	1.76	WH	Newly discovered	WRKY
osa-miR396e-5p_1ss19CT	TRITD_5Bv1G146000-2	probable WRKY transcription factor 2	−2.32	2.04	WH	Newly discovered	
tae-miR398	TRITD_7Av1G028020	putative WRKY transcription factor 72	−inf	1.74	HS	Newly discovered	
osa-miR444b.1_1ss21CT	TRITD_3Av1G213210	putative WRKY transcription factor 33	−1.32	2.58	HS	Newly discovered	
tae-MIR5048-p3_2ss18TG21GA	TRITD_3Av1G074690-4	WRKY transcription factor, partial	−0.66	1.63	HS	Newly discovered	
ata-MIR9674b-p5	TRITD_3Av1G213210	putative WRKY transcription factor 33	−3.53	2.58	HS	Newly discovered	
	TRITD_3Bv1G195790	WRKY27 transcription factor	−3.53	1.88	HS	Newly discovered	
osa-miR530-5p_L+1_1ss21AT	TRITD_5Av1G046770	MYB ^1^ transcription factor 79	−inf	4.00	HS	Newly discovered	MYB
ata-miR5168-5p	TRITD_5Av1G118710	transcription factor MYB44-like	−inf	4.57	HS	Newly discovered	
			−inf	2.67	WH	Newly discovered	
tae-MIR9772-p5	TRITD_5Av1G207760-4	MYB-related protein	−0.66	2.95	HS	Newly discovered	
tae-miR408_L-1	TRITD_6Av1G199570-2	bZIP ^1^ transcription factor	−3.55	3.69	HS	Newly discovered	bZIP
tae-MIR5384-p5	TRITD_3Bv1G212830	bZIP domain containing protein	−1.93	2.98	HS	Newly discovered	
			−1.98	1.97	WH	Newly discovered	
bdi-MIR530b-p3_1ss14GC	TRITD_3Av1G067970	bHLH13-like ^1^	−inf	2.14	HS	Newly discovered	bHLH
tae-MIR9662b-p5_1ss9CG	TRITD_2Av1G047140	bHLH47	−1.66	2.42	HS	Newly discovered	

^1^ DREB, dehydration-responsive element-binding protein. NAC, NAC (NAM, ATAF1/2, and CUC2) domain containing transcription factors. ARF, auxin response factors. WRKY, transcription factors with the WRKY domain containing the highly conserved amino acid sequence WRKYGQK. MYB, MYB (myeloblastosis) family of transcription factors. bZIP, basic-leucine zipper (bZIP) transcription factor family. bHLH, basic/helix–loop–helix (bHLH) transcription factors. ^2^ −inf, represents infinite value of a negative fold-change (as the miRNA expression level under stress was zero).

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
