# Peer review of "Integrated Analysis of Small RNA, Transcriptome, and Degradome Sequencing Reveals the Water-Deficit and Heat Stress Response Network in Durum Wheat"

_ijms, 2020, doi:10.3390/ijms21176017_

Round 1

Reviewer 1 Report

Review_ ijms-889976

In this manuscript (ID: ijms-889976), authors provided a comprehensive description of miRNA and miRNA-mRNA regulatory modules in durum wheat under water-deficient and/or heat stresses. NGS approach was applied to durum for the first time and huge data sets were well analyzed. However, please consider comments below to improve the manuscript;

  • Although durum genome was analyzed with NGS for the first time in the present study, previously with different methods miRNA and miRNA-mRNA relation was investigated in durum (references 30-36) including miRNA from durum wheat under water deficiency reported by authors. Does present NGC data cover all previous data? How many new miRNAs was revealed? Is there no discrepancy between present and previous data? What is unique and novel in the present data apart from the technical/methodological differences? Table 1 indicates several “newly discovered” paring-type but this is “example”. Please mention clearly how many “newly” discover among total detected pairs that include previously reported. In addition, importance of pairs appearing in Table 1 in water-deficient and/or heat stresses is unclear.

  • It is described in the results section that “miRNAs and their targets do not always exhibit negative correlation in their expression. As the regulatory relationships of miRNA-target modules are not always one-on-one specific, one miRNA can regulate multiple mRNAs and one mRNA can be synergistically targeted by several miRNAs.” (lines 369-372). Also “The miRNA-guided stress responsive networks are multi-layered, where one miRNA can regulate multiple mRNAs and one mRNA can be synergistically targeted by several miRNAs” (lines 519-520). These sound reasonable, but tell us almost no biological means. How such network can make plants stress-tolerance? How summarized interactions (Fig9b) explain the different tolerance between DBA Aurora and L6? I agree miRNA-mRNA paring identified in the present study with NGS approach. However, analysis and identified most important miRNA-mRNA regulatory module(s) should be essential and be detected to enhance tolerance to certain stresses as discuss in page 9. General scheme or no-weighted miRNA-mRNA network like Fig9b is hardly “valuable attributes for building stress tolerance” (Line 502). TFs may control down-stream components for stress tolerance (as authors mentioned as novel pairs of miRNAs-TF). If so some special TA (and associated miRNA) can be high-lighted and such key pair(s) can be analyzed intensively as Fig.8. In the present Fig.8, TFs and down-stream components were equally treated.

Small point

  • Why Y-axis categoly of Fig 4 (GO) and Fig 6 (KEGG) are out of order in each panel? In the present style it is very difficult to compare each other.
  • In Fig9a, in second block (Signal perception), “bdi-miR159a-5p_R+1” should be red, and “calcium-transporting ATPasae 8” should be green?

Reviewer 2 Report

Title: Integrated analysis of small RNA, transcriptome and degradome sequencing reveals the water-deficit and heat stress response network in durum wheat

Confidential Comments to the EIC:

This study focused on the molecular mechanism of durum wheat in response to water-deficit and heat stress by integrating analysis of small RNA, transcriptome and degradome. The results were clearly presented and novel results was obtained. However, the major results were not properly discussed in the discussion section. The introduction and discussion should be carefully improved. After the required revisions, I will recommend publishing. Major point: why the time point of water deficit stress treatment was set at the pre-anthesis, while the time point of heat stress treatment was post-anthesis?

Comments to the author:

Abstract: the key points were highlighted, while several sentences need to be rephrased.

L15-16 Please rephrase this sentence.

L17-19 Please rephrase this sentence.

L21 delete “pathways”

Introduction:

This section should be carefully improved. The physiological and molecular mechanisms (miRNAs) of water-deficit and heat stress should be summarized, especially in wheat. In addition, some sentences need to be improved.

L35-36 Please rephrase this sentence, it may make readers confusing.

L39-40 Please rephrase this sentence.

L40-42 Please rephrase this sentence, it may make readers confusing.

L43-45 Please rephrase this sentence, it may make readers confusing.

L84-104 Maybe this section can be removed.

Materials and methods

This section is fine, the details of experimental setup and methods has been well described, while the details of qPCR and statistical analysis should be added.

Results

This section is fine, the main results has been presented clearly. However, there are some errors needed to be corrected.

L136-137 Please rephrase this sentence.

L140-141 Please rephrase this sentence.

L245 changes “cell surface” to other words

L310-311 Please rephrase this sentence.

Discussion

This section should be carefully improved. The point “key miRNA-mRNA modules with antagonistic regulatory patterns subject to different stresses” needs to be verified by experiments or references. In this study, the author obtained this point by qPCR experiment, which is unreasonable. In addition, more related references should be added.

L418-419 Please rephrase this sentence, it may make readers confusing.

L454 changes “conferred” to “enhanced”

L454 delete “high”

L455-457 references should be added

L489 delete “stress”

L538-539 Please rephrase this sentence.

L567 changes “reduction” to “degradation”

In addition, changes “chloroplastic” to “chloroplast”.

Other comments: 1) changes “durum” to “durum wheat”.

2) in the WH (water-deficit plus heat stress) group, the time point of stress treatment is both pre-anthesis or both post-anthesis or pre-anthesis (water-deficit) and post-anthesis (heat stress)?

3) what is the main purpose of water-deficit plus heat stress treatment?

Reviewer 3 Report

I think the results in this paper are strong, but their presentation needs a lot of work in order to make this paper accessible for the general audience. Most of my comments pertain to the main figures.

Major Comments:

1) A summary table at the beginning of the results highlighting the main abbreviations and time points would be very useful to help the reader remember what all of the abbreviations and symbols mean.

2) The main figures need to be majorly reworked. Your main figures should only show the conclusions most important for your paper. Right now, they are filled with way too much information for the reader to digest. Below I provide specific comments for each figure.

3) Figure 1: show only the venn diagrams that highlight the comparisons in the previous paragraph. Limit venn diagrams to 2 or 3 comparisons if possible - the 5 comparisons together is very hard to read. 

4) Figure 2: Font size is much too small to read right now, and it is unclear which heatmap is using which comparison. Consider using less DEMs - perhaps for the heatmap you could use a more stringent p-value so that the pattern is more visible.

5) Figure 3 should move to supplement. This is not a main conclusion of the paper.

6) Figure 4: Same comment as Figure 2 - condense your results to the main comparisons so that we can read the font and interpret your conclusions.

7) Figure 5 should move to supplement. This is not a main conclusion of the paper.

8) Figure 6: This is very similar/complementary to Figure 4. Consider combining them in some way. 

9) Figure 7: Individual panels need titles and font size needs to be increased.

10) Figure 8: This figure is fine, although more separation between panels a and b would be nice (right now hard to see where they begin and end)

11) Figure 9: This figure is fine, larger font size would be nice

Minor comments

1) How was the network in Figure 9B generated? I was not able to find this detail in the text.

2) The volcano plots in Figure S4 are not very informative. Usually volcano plots have a much larger spread of points than this. Please try using another plotting function/device to better show the distribution of the data points, or perhaps do not include them in the final version of the manuscript.

3) I think section 2.3 does not need to be its own section in the results and can be combined with section 2.4. 

Round 2

Reviewer 1 Report

Most parts are adequately revised in the new version. However, Figure 3 (previous Figure 6) is not yet improved well.

  • In L6 of Fig6a, “Metabolism of Cofactors and Vitamins” is placed 3rd position from the top, but 9th position in Fig6b and others.
  • In DBA Aurora of Fig6a, top “Environmental Adaptation” classified as Organismal systems should be blue-colored as other panels.
  • In DBA Aurora of Fig6a, 7th “Fording, Sorting, and degradation” (Genetic information processing) should be yellow-green as others.
  • In DBA Aurora of Fig6a, color of Class “Metabolism” including bars 2nd to 6th should be changed according to other panels.

Reviewer 2 Report

The author answered the major points in this revised manuscript. The physiological and molecular mechanisms (miRNAs) of water-deficit and heat stress were summarized in wheat in the introduction. In the materials and methods section, the details of qPCR was also added. In addition, the point of “key miRNA-mRNA modules with antagonistic regulatory patterns subject to different stresses” was discussed in detail by citing more references. Thus, I recommend publishing the manuscript as it meets all the requirements after revised.

Author Response

Response: We thank Reviewer 2 for the positive comment.